# Matrix Mixture of Experts is the Best Fast Feed-Forward

**Vyacheslav Chaunin**
Tomsk State University
Tomsk, Russia
`slava.chaunin@gmail.com`

**Nikolay Mikhaylovskiy**
Tomsk State University & NTR
Tomsk, Russia
`nickm@ntrlab.com`

## Abstract

We dissect the recently introduced Fast Feed-Forward (FFF) neural network architecture and propose a matrix formulation of FFF that allows a unified perspective on FFF and Mixture of Experts (MoE) architectures. This formulation achieves, on average, nearly a 4x speedup in FFF inference on CPUs and 12x on GPUs, compared to the original formulation, for FFF depths up to 8. Furthermore, we demonstrate that this formulation allows for modifications to the activation function. Specifically, we observe that using a linear activation yields slightly improved performance for deeper layers while maintaining equivalent training time.

## 1 Introduction

Recent years have witnessed significant progress in the field of machine learning, marked by the development of increasingly complex and powerful models. One of the promising directions in this domain is the application of *Mixture of Experts* (MoE) methods (Jacobs et al., 1991; Fedus et al., 2022b;a; Jiang et al., 2024). At the time of writing, the leading open-source LLM uses MoE (DeepSeek-AI et al., 2025).

The motivation behind employing MoE lies in addressing the challenges of scalability and efficiency in large neural networks. Modern machine learning tasks often require substantial computational resources. MoE techniques enable efficient allocation of computational resources at inference time by activating only those submodels that are most relevant for a given task. This approach reduces the overall computational overhead while maintaining high-quality predictions.

MoE was first introduced in *"Adaptive Mixtures of Local Experts"* by Jacobs et al. (1991). The central idea is a system comprising multiple models, each specializing in a subset of training data. Alongside these experts, a gating model is trained to determine the weights of each expert during inference.

Over time, the MoE concept has evolved, finding applications within the layers of deep neural networks, including large-scale language models (Shazeer et al., 2017; Fedus et al., 2022b; Gale et al., 2022; Jiang et al., 2024; DeepSeek-AI et al., 2025). A version of MoE, *Hierarchical Mixture of Experts* (HMoE, first introduced by Jordan & Jacobs (1994)), dubbed *Fast Feed-Forward Networks* (FFF) in a namesake work by Belcak & Wattenhofer (2023b) remains less explored than traditional MoE methods.

In this work, we present a matrix-based formulation of the FFF architecture that offers a unified parameterization with MoE models. This formulation results in a significant enhancement in efficiency, achieving nearly 4x faster inference on CPUs for networks with depths up to 8, and 9x faster on GPUs for depths up to 13. Additionally, we demonstrate that this formulation enables modifications to the inner activation inside FFF. Notably, we find that the use of linear activation slightly improves performance for deeper layers while maintaining the same training time.

## 1.1 Feed-forward layers

Fully connected (feed-forward) layers are foundational to modern neural network architectures. In practice, these layers are typically expressed as:

$$\mathbf{z} = f(\mathbf{Wx}),$$

where $f$ denotes the activation function, and $\mathbf{W}$ represents the weight matrix.

The computational complexity of such layers is proportional to the number of parameters $N$ and can be expressed as $\mathcal{O}(N)$. These dense connections provide high flexibility, enabling models to capture intricate dependencies in data.

However, as the size and depth of the network increase, fully connected layers become increasingly resource-intensive. In particular, during inference, only a subset of neurons is activated, as most input data is sparse. This means that the features relevant to a specific instance are far fewer than the total number of features the model can represent, leaving many neurons inactive. This inefficiency is particularly pronounced in large-scale architectures, such as language models, where a substantial proportion of parameters are concentrated in fully connected layers.

These challenges naturally motivate the development of models that maintain the expressivity of fully connected layers while reducing computational resource requirements. One promising direction in this context is the Mixture of Experts architecture, which addresses these limitations by sparsely activating subsets of parameters.

## 1.2 Mixture of Experts (MoE)

The primary concept of the MoE architecture involves partitioning a neural network layer (primarily fully connected) into independent blocks of neurons, called *experts*. A routing layer, also known as a *router*, precedes the experts and determines the subset of experts to be utilized at the inference time. The performance of the model is highly sensitive to the choice of router. Recently, the most common mechanism for selecting experts involves calculating selection probabilities and using a weighted average over the selected experts (also see Figure 1):

$$R(i|\mathbf{x}) = \text{Softmax}(W\mathbf{x})_i \tag{1}$$

$$\mathbf{y} = \sum_{i \in \text{top-}k(R)} R(i|\mathbf{x}) f_i(\mathbf{x}) \tag{2}$$

where $R(i|\mathbf{x})$ represents the probability of selecting the $i$-th expert, which is represented by the layer $f_i$.

For a MoE layer containing an equivalent total number of parameters as a dense fully connected layer, inference is faster because only a subset of experts is activated. The number of operations required during inference can be expressed as:

$$\mathcal{O}(K + E \times k)$$

where $K$ is the total number of experts, $E = \frac{N}{K}$ is the size of each expert, and $k$ is the number of experts used during inference.

## 1.3 Fast Feed-forward (FFF)

Belcak & Wattenhofer (2023b;a) introduced the FFF layer architecture. The primary idea is to achieve exponential acceleration during inference by selecting an expert from an exponentially large set and replacing the MoE router layer with a binary tree structure (see Figure 2).

Each node $i \in [1, 2^d - 1]$ in the tree contains a parameter vector $\mathbf{w}_i \in \mathbb{R}^n$. Based on the dot product between the input vector and the node's parameter vector, the probability of selecting the next parameter vector for one of the child nodes is computed as:

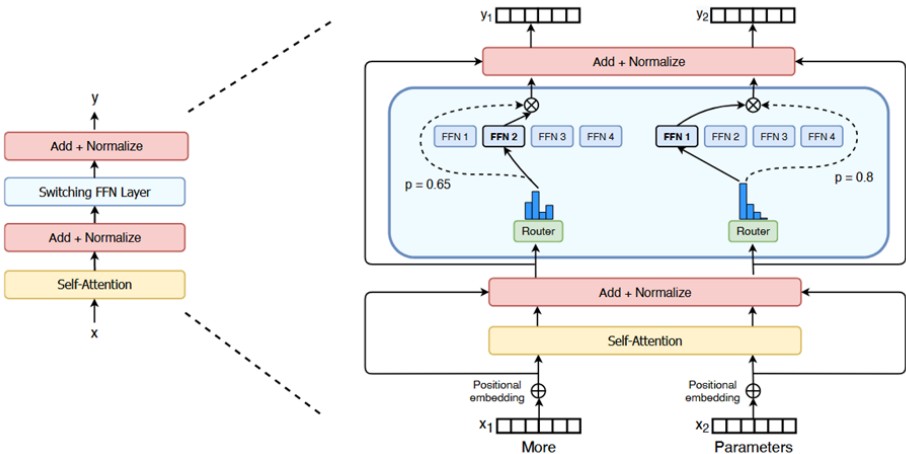

Figure 1: Diagram of MoE layer from Fedus et al. (2022b)

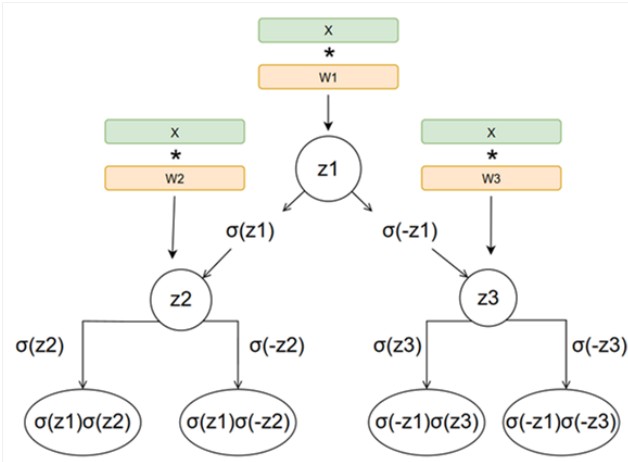

Figure 2: A diagram of a depth-2 tree in the FFF layer. The final layer provides probabilities for the leaves.

$$P(2i|\mathbf{x}, i) = \sigma(z_i) = \sigma(\mathbf{w}_i^\top \mathbf{x}) \tag{3}$$

where $\sigma(x) = \frac{1}{1+e^{-x}}$. The probability of selecting the opposite child node is given by:

$$P(2i+1|\mathbf{x}, i) = \sigma(-z_i) = 1 - \sigma(\mathbf{w}_i^\top \mathbf{x}) \tag{4}$$

At the final level of the tree are the leaf nodes in , similar to the experts in MoE, henceforth referred to as experts. During training, data passes through all tree nodes, accumulating probabilities to form a distribution over the leaf nodes at the final layer. These probabilities are subsequently used to compute the weighted average:

$$\mathbb{E}_{i \sim R(i|\mathbf{x})}[f_i(\mathbf{x})] \tag{5}$$

where $R(i|\mathbf{x})$ denotes the probability of selecting leaf $i$, which is represented by the layer $f_i$.

During training, an FFF layer of depth $d = \log K$ uses all $\mathcal{O}(2^d \times E + 2^{d-1})$ parameters to compute the average, ensuring all experts are utilized. This approach enables a differentiable router, facilitat-

ing accurate gradient computation. However, it significantly increases computational costs during training (Muqeeth et al., 2023).

During inference, the FFF layer sequentially selects the most probable node at each level of the tree in a greedy manner, with the output of the selected leaf node serving as the final output vector. The computational complexity of this algorithm is logarithmic in the number of leaves, given by:

$$\mathcal{O}(d + E)$$

Thus, for an equivalent number of experts, FFF achieves exponential acceleration in search. However, this algorithm lacks parallelism, as each tree level must be computed sequentially.

## 2 COMMON PARAMETRIZATION FOR FFF AND MOE

### 2.1 LOGARITHMIC TRANSFORMATION

Consider the router in FFF, which represents the distribution of the selected experts $R(i|\mathbf{x})$. It is defined as follows:

$$R(i|\mathbf{x}) = \prod_{j \in \text{Path}(i)} P(j|\mathbf{x}, \lfloor \frac{j}{2} \rfloor) \tag{6}$$

where $\text{Path}(i)$ contains the indices of all nodes along the path from the root to the leaf $i$, and the probabilities $P$ inside the product take the form of equations 3 or 4 depending on the parity of $j$, parameterized by weight vectors.

The logarithm of this distribution can be computed as:

$$\log R(i|\mathbf{x}) = \sum_{j \in \text{Path}(i)} a((-1)^j z_{\lfloor j/2 \rfloor}) \tag{7}$$

where $a(x)$ is defined as:

$$a(x) = \log \sigma(x) = -\text{Softplus}(-x) \tag{8}$$

In this equation, $(-1)^j$ indicates that for odd nodes, the probability is computed as the activation of an opposite value, as shown in equation 4. Expression in 7 represents a linear combination of activations from a subset of tree nodes. It can also be interpreted as a weighted sum of activations from all nodes[1], similar to the pre-activation in the next layer of a fully connected network for layers of size $2^d - 1$ and $2^d$.

### 2.2 LINEARITY

Since the expression 7 is a linear combination, it can be represented in a matrix form. We can also convert this expression into probabilities by normalization. If $\mathbf{z} = (z_1, \ldots, z_{2^d-1}) = W\mathbf{x}$, the expert distribution is expressed as:

$$R(i|\mathbf{x}) = \text{Softmax}(Ta(S\mathbf{z}))_i \tag{9}$$

where $T \in \mathbb{R}^{2^d \times 2(2^d-1)}$ is a matrix where $t_{ij} = 1$ if $j \in \text{Path}(i)$ and 0 otherwise, and $S \in \mathbb{R}^{2(2^d-1) \times 2^d-1}$ contains vertical $\pm 1$ pairs along the diagonal to account for the sign before the activation or more formally:

$$S_{ij} = \begin{cases} 1 & \text{if } 2i - 1 = j \\ -1 & \text{if } 2i = j \\ 0 & \text{otherwise} \end{cases}$$

---

[1]The weights are 1 for the nodes on the path to the given leaf and 0 for all others.

For example if FFF layer has depth $d = 2$, then $T$ and $S$ are constructed as follows:

$$T = \begin{pmatrix} 1 & 0 & 1 & 0 & 0 & 0 \\ 1 & 0 & 0 & 1 & 0 & 0 \\ 0 & 1 & 0 & 0 & 1 & 0 \\ 0 & 1 & 0 & 0 & 0 & 1 \end{pmatrix} \quad S = \begin{pmatrix} 1 & 0 & 0 \\ -1 & 0 & 0 \\ 0 & 1 & 0 \\ 0 & -1 & 0 \\ 0 & 0 & 1 \\ 0 & 0 & -1 \end{pmatrix}$$

Firstly, this formulation illustrates that the FFF architecture is a specific instance of a more general parameterization, given by $Ta(S\mathbf{z})$, where $T$, $S$, and $a$ can be chosen arbitrarily. For example, the MoE model can be recovered by setting $S = T = I$ (the identity matrix) and defining $a(x) = x$.

Furthermore, the FFF architecture can be understood as computing the distribution over $2^d - 1$ experts, followed by an inductive bias that regroups and filters this distribution to produce probabilities for $2^d$ experts. This perspective highlights how the architecture leverages intermediate computations to enable more expressive modeling with minimal structural changes.

Secondly, this formulation clarifies the potential for parallelizing computations of the probabilities. However, a possible limitation of this approach is the challenge of storing these matrices in memory, especially when represented in their full form. Despite this, it is evident that the matrices are highly sparse, which allows for efficient storage using specialized data structures, such as the Compressed Sparse Row (CSR) format, to significantly reduce memory overhead while maintaining computational efficiency.

## 2.3 CHANGING THE ACTIVATION FUNCTION

Let us examine the activation function in the original FFF architecture in detail. By removing the sign from the argument and flipping the signs of all weights at the nodes, we can construct an equivalent model. Additionally, the second sign in the Softplus function serves a critical purpose: it ensures that, during the transition to probabilities, the resulting values lie within the interval $[0, 1]$. However, this second sign can be removed if appropriate normalization is applied at the end of the computation to achieve the same effect.

Given these considerations, we adopt the standard form of the activation function, $a(x) = \text{Softplus}(x)$, as the default choice for the base FFF architecture.

With the flexibility to adjust our current parameterization, we further experiment with alternative activation functions within the model to explore their impact on performance.

## 3 EXPERIMENTS

To assess the impact of activation functions on the FFF architecture, we conducted experiments on small FFF-only models on synthetic and CIFAR-10 Krizhevsky & Hinton (2009) datasets, which was chosen due to its alignment with the FFF paper and for rapid iteration in numerous model configurations. We trained several FFF models with varying depths, ranging from 2 to 5. For activation functions, we selected the following options: the baseline $\text{Softplus}(x)$, a linear activation $f(x) = x$, the ReLU function $\text{ReLU}(x) = \max(0, x)$, and the GELU function $\text{GELU}(x) = x \cdot \Phi(x)$. For comparison, we also trained MoE models of equivalent size by setting the number of experts equal to that in FFF, i.e. $2^d$, while leaving other parts of the model unchanged.

## 3.1 SYNTHETIC DATA AND CIFAR-10

The model architecture employed in this setup consists of a single FFF, with experts represented as two-layer MLPs that contain 8, 16, and 32 hidden neurons, respectively.

For the synthetic dataset, we generated random clusters of isotropic Gaussian blobs, augmented with additional features $x^2$ and $\sin(x)$, both of which included noise. The dataset consists of four distinct clusters, each containing six-dimensional data points, with labels assigned according to their respective clusters.

For CIFAR-10 each model was trained for 20 epochs with a batch size of 64, using the Adam optimizer and a one-cycle learning rate schedule. The maximum learning rate was set to $8 \times 10^{-4}$ to facilitate faster convergence. To prevent overfitting, early stopping was applied, and dropout with a rate of 0.2 was used before each expert. For the synthetic dataset, the batch size was adjusted to 16, and the number of epochs was reduced to 15.

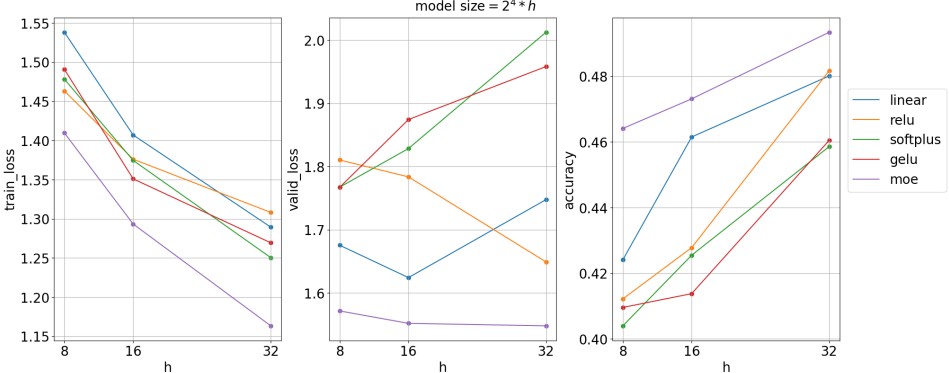

Figure 3: CIFAR-10 loss on training and validation datasets, and validation accuracy for FFF models with depth 4 and MoE model with 16 experts.

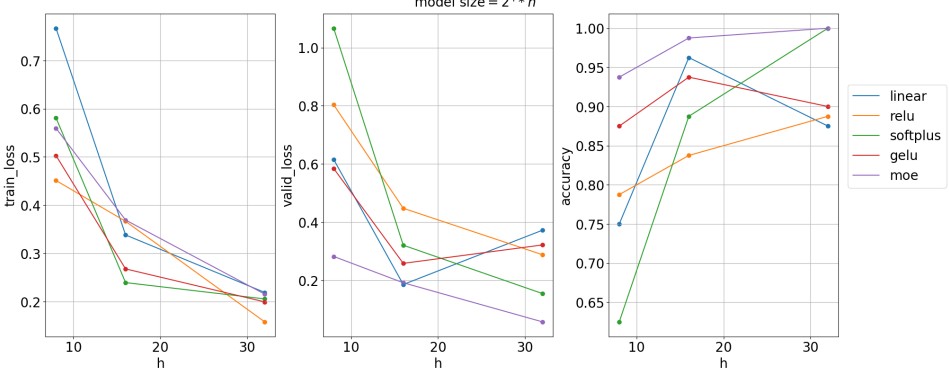

Figure 4: Losses on the syntetic data

The results are similar for both datasets and demonstrate that the validation loss is consistently lower for the linear activation function in the FFF model compared to other activation functions, despite all models being trained for the same number of iterations. This suggests that the linear activation function generalizes more effectively while maintaining the same model size and depth. Notably, this effect is less pronounced in shallower models, likely because there is sufficient signal propagation when the model's depth is limited. Figure 3 presents the accuracy and loss trends for models of depth 4, showcasing the performance dynamics across model widths and activation functions. Additional plots for models with other depths can be found in the Appendix B.

The average change in relative model accuracy was analyzed for different activation functions. The baseline configuration, an FFF model with Softplus activation, represents the activation used in the original formulation. The average change in accuracy, $\Delta A_{\text{avg}}$, was calculated across various model depths. Notably, the ReLU activation function resulted in a modest accuracy improvement of 0.76%, while the linear activation function achieved a slightly higher improvement of 3.34%. These results are summarized in Table 1.

The decrease in performance observed when activation functions are incorporated into the model is likely attributed to the following factors: the activation functions tend to obstruct the signal flow from certain features, which negatively affect the router probabilities. This interference complicates the model's ability to adapt effectively within the FFF architecture, particularly as the depth of

the model increases. Consequently, models with activation functions exhibit inferior performance, especially in deeper FFF layers.

| Activation | d=2.0 | d=3.0 | d=4.0 | d=5.0 |
|---|---|---|---|---|
| Linear | **0.470** | 0.461 | **0.455** | **0.447** |
| ReLU | 0.464 | **0.468** | 0.441 | 0.416 |
| Softplus | 0.458 | 0.461 | 0.429 | 0.427 |
| GELU | 0.461 | 0.456 | 0.428 | 0.408 |
| MOE | **0.472** | **0.483** | **0.477** | **0.470** |

Table 1: Accuracy values for different activation functions across varying $d$ values of the FFF model on CIFAR-10

## 3.2 CRAMMING BERT

Additionally, we compared activation functions by training FFF layers within a smaller transformer model. For the baseline, we utilized the "Cramming BERT" setup (Geiping & Goldstein, 2022), as employed by the authors of the FFF paper. The experimental setup follows the default configuration used in Cramming but with a slightly reduced model size (see Appendix A for details).

| Activation | STSB | SST2 | RTE | MRPC | COLA | Average GLUE |
|---|---|---|---|---|---|---|
| Linear | **0.564** | 0.859 | 0.484 | 0.706 | 0.129 | 0.57 |
| Softplus | 0.439 | **0.860** | 0.509 | 0.708 | 0.112 | 0.54 |
| ReLU | 0.560 | 0.849 | **0.560** | 0.716 | 0.117 | **0.58** |
| GELU | 0.506 | 0.855 | 0.545 | **0.723** | **0.165** | **0.58** |

Table 2: Performance of different activations on GLUE tasks.

We selected an FFF depth of 3 as the baseline configuration, representing a moderate depth that balances complexity and performance. The results, summarized in Table 2, demonstrate the performance of each configuration on the GLUE benchmark. Although the observed deviations were less pronounced — probably because the attention mechanism is the dominant contributor to transformer performance — slight improvements over the Softplus baseline were still noticeable. However, attributing these improvements to the choice of activation function is challenging, as the differences may be within the noise margin. Further experiments with larger FFF depths could provide more definitive insights and represent a promising direction for future research.

## 4 BENCHMARKS

Benchmarking was conducted to evaluate the computational performance of the FFF and MoE layers in different implementations. Since the calculation of expert outputs is independent of the method, the focus was on comparing the speed of probability computation for all routers. The tests included three variants of FFF and one MoE implementation, all of which have nearly identical numbers of parameters. The number of parameters is given by the expression $2^d \cdot n_{\text{input}}$ for MoE and $(2^d - 1) \cdot n_{\text{input}}$ for FFF implementations, where $n_{\text{input}}$ denotes the input vector dimension.

The "**FFF orig**" implementation replicates the original training-mode FFF, sequentially computing probabilities layer by layer. "**FFF inference**" modifies this for inference, processing only one branch per layer based on the sign of a single dot-product. "**FFF logs**" operates similarly to the original but uses logarithmic transformation 7 and applies exponentiation only at the last step. "**FFF T, S**" optimizes the process using three matrix multiplications in dense configurations as described in 9.

The "**MoE**" implementation follows the standard Mixture of Experts structure, computing Softmax($W\mathbf{x}$) via a single matrix multiplication, i.e. with $S = T = I$, eliminating the extra computation step. These variants were compared for computational efficiency and scalability.

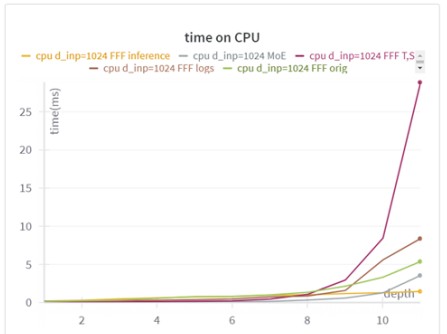 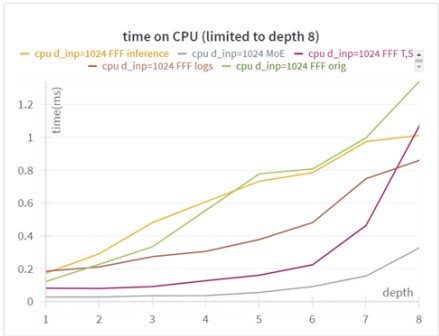

Figure 5: Performance comparison of the FFF layer in different implementations and MoE on CPU

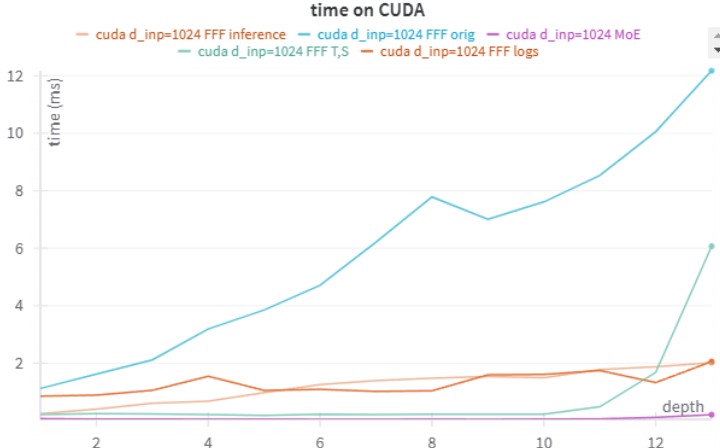

Figure 6: Performance comparison of the FFF layer in different implementations and MoE on GPU

| Method | GPU | | CPU | |
|---|---|---|---|---|
| | Up to 8 | Up to 13 | Up to 8 | Up to 13 |
| MoE | **45.63** | **57.84** | **7.27** | **4.55** |
| FFF T,S | 11.89 | 8.93 | 3.55 | 0.88 |
| FFF Inference | 4.22 | 4.52 | 0.90 | 1.09 |
| FFF Logs | 2.58 | 3.22 | 1.27 | 1.07 |
| FFF Orig | 1.00 | 1.00 | 1.00 | 1.00 |

Table 3: Average time ratios relative to original implementation with the same number of parameters

## 4.1 AVERAGE SPEEDUP

The average speedup was computed using the harmonic mean of the time ratios between **"FFF T,S"** and **"FFF orig"** across varying depths, as shown in Table 3. The results demonstrated a 3.5x speedup on CPUs for depths up to 8, corresponding to 256 experts, a scale that is already considerable and in line with common practices where the number of experts typically does not exceed a few hundred. Beyond this depth, performance deteriorated exponentially due to the computational overhead of dense matrix operations. On GPUs, as shown in Figure 6, **"FFF T,S"** achieved a 9x speedup over **"FFF orig"**, while the classical **"MoE"** model was 4.9x faster. This disparity is attributed to the **MoE**'s direct computation of probabilities following the first matrix multiplication, whereas **FFF** requires an additional step of distributing the probabilities across experts. Despite the observed improvements, the results underscore the trade-off between computational efficiency and architectural complexity.

## 5 CONCLUSION

This work presents a matrix-based formulation of the FFF architecture, providing a unified view with MoE models. The approach achieves up to 4x and 9x faster inference on CPUs and GPUs for depths up to 8, while improving accuracy on CIFAR-10 by about 3%, driven by changes in the activation function. Benchmarking shows that FFF T,S outperforms the original FFF but remains 6.4x slower than MoE on GPUs. FFF is efficient with more than $2^8$ experts on CPU or $2^{13}$ on GPU; otherwise, MoE or the matrix formulation is preferable. These results highlight the scalability and efficiency of the proposed formulation.

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

## A  CRAMMING EXPERIMENTAL PARAMETERS

The cramming BERT model was trained with the following configuration: it utilized 8 transformer layers and 8 attention heads, with a model size of 512. The hidden size in the expert MLP was set to 64, and the FFF component had a depth of 3. Training was conducted with a batch size of 4096 over 50,000 pretraining iterations.

## B  ADDITIONAL RESULTS FOR MODELS WITH VARYING DEPTH

This section presents the accuracy and loss plots for shallow FFF models with depths of 2 and 3, as well as an FFF model with a depth of 5, corresponding to the experimental setup described in Section 3.

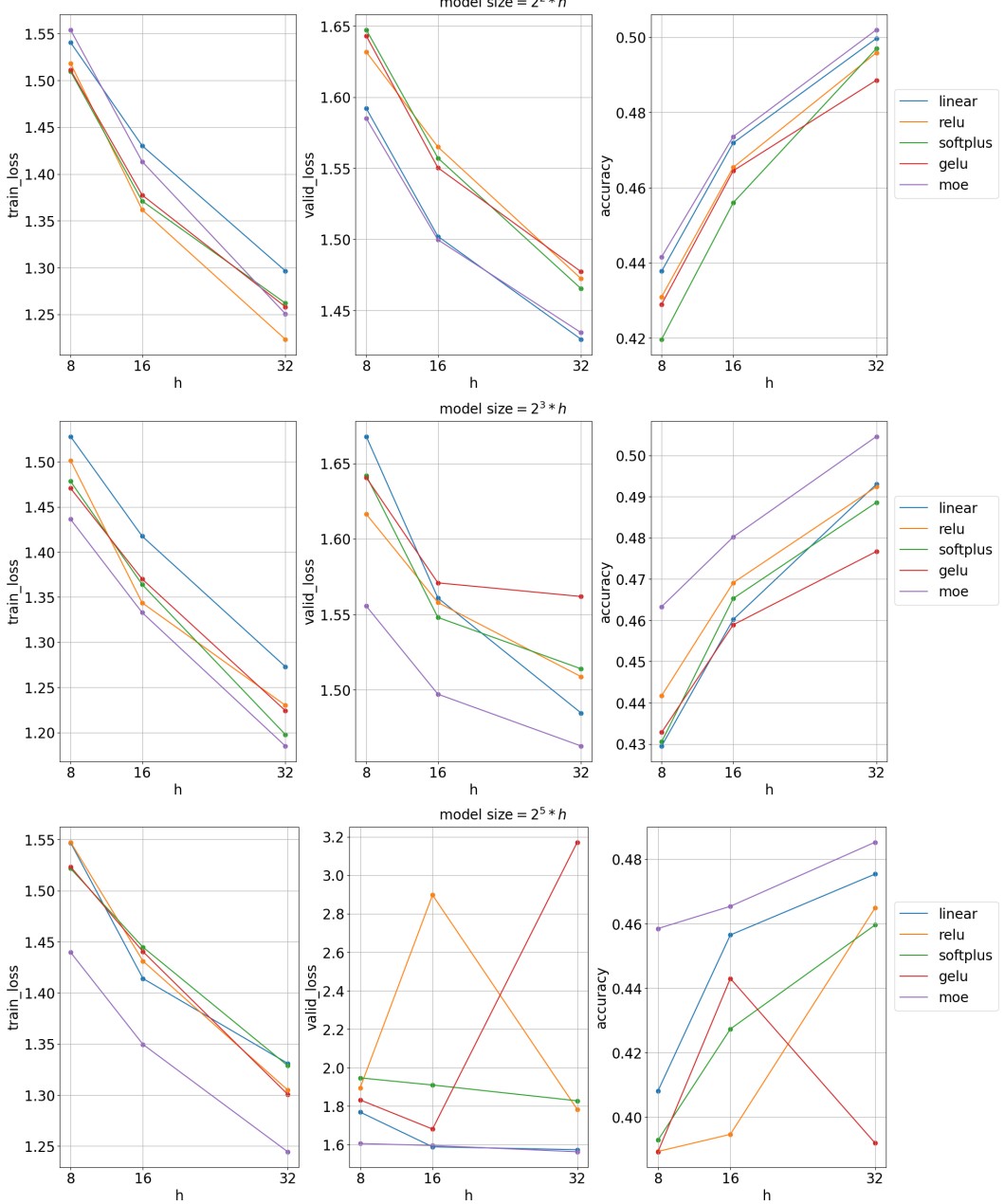

Figure 7: Accuracy and loss plots for models with depths of 2, 3, and 5 on CIFAR-10. MoE slightly outperforms any FFF with a similar configuration.

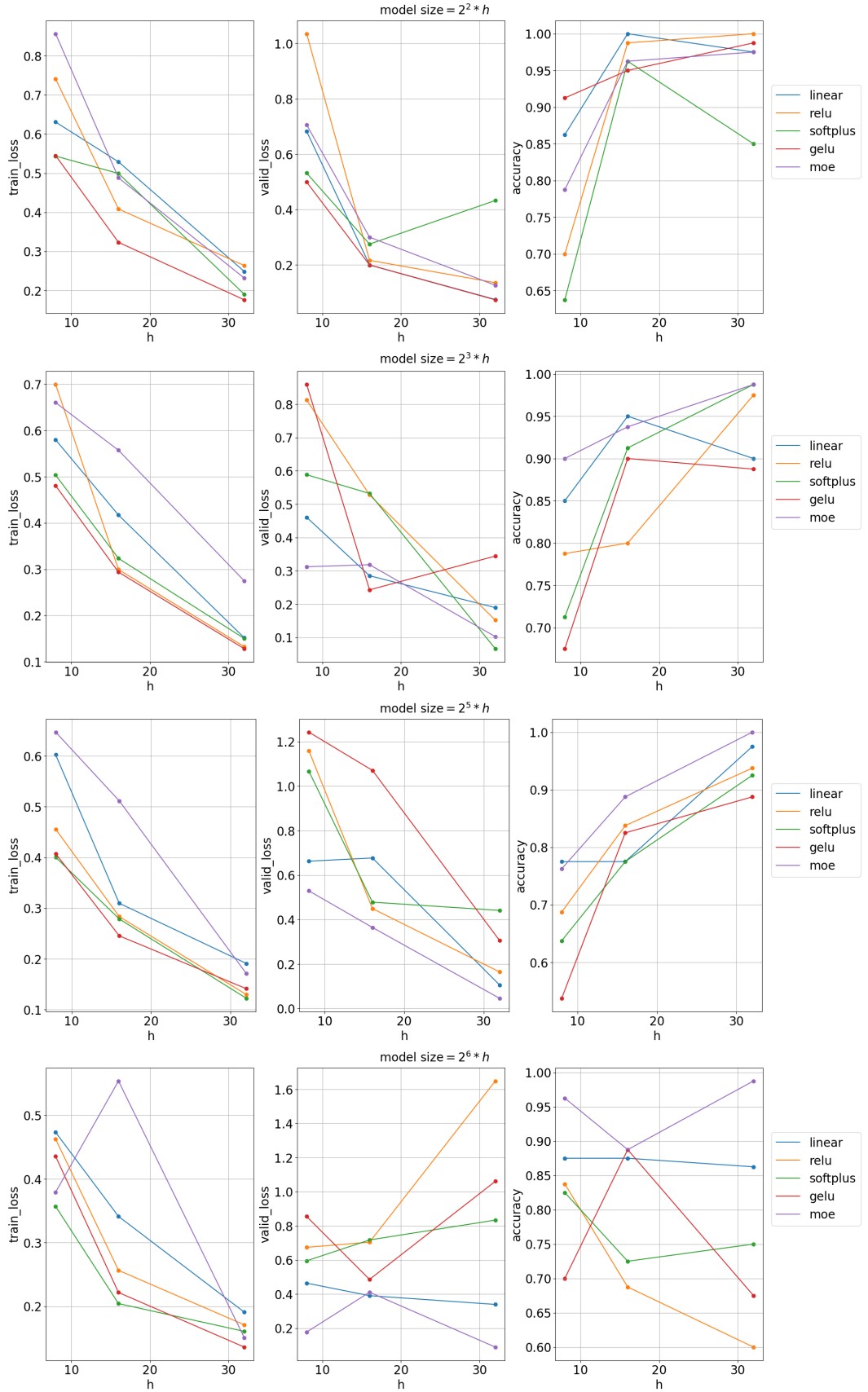

Figure 8: Accuracy and loss plots for models with depths of 2, 3, 5 and 6 on the synthetic dataset.

