# OpenReview forum: "Matrix Mixture of Experts is the Best Fast Feed-Forward"
_mathai.club/MathAI/2025/Conference — MathAI 2025 Oral_

### Official Review · Reviewer_W5nu · 2025-02-24
**MATRIX MIXTURE OF EXPERTS IS THE BEST FAST FEED-FORWARD**

**Rating:** 8
**Confidence:** 3

**Review:**

This work presents a matrix-based formulation of the Fast Feed-Forward (FFF) neural network architecture, providing a unified view
with Mixture-of-Experts (MoE) models. The approach achieves up to 4 times and 9 times faster inference on CPUs and GPUs for
depths up to 8.  Experiments on well-known benchmarks show that the FFF variant that optimizes the process using three matrix multiplications in dense configurations, outperforms the original FFF but remains 6.4 times slower than MoE on GPUs. FFF is efficient with more than 28 experts on CPU or 213 on GPU; otherwise, MoE or the matrix formulation is preferable. These results highlight the scalability and efficiency of the formulation proposed by the authors.

I have only one minor remark. It seems that the comparative results with three different activation functions are of preliminary kind. It would be helpful if the authors gave some ideas explaining the reasons why one activation function performs better than another. It is interesting to see the results in Figures 2 anf 5 for larger number of layers and for different datasets. Would they lead to the same conclusions?

---

### Official Review · Reviewer_7VYd · 2025-02-27
**The paper presents an interesting and theoretically sound approach to unifying FFF and MoE, with notable improvements in speed and accuracy. However, the limited scope of experiments and lack of comparison with full-scale architectures reduce its practical impact.**

**Rating:** 6
**Confidence:** 4

**Review:**

Review of "Matrix Mixture of Experts is the Best Fast Feed-Forward"
Relevance:
The paper presents a novel matrix-based formulation of the Fast Feed-Forward (FFF) architecture, unifying it with Mixture of Experts (MoE). This approach addresses scalability and efficiency challenges in large neural networks, particularly for inference tasks. The work is timely, given the growing demand for efficient deep learning models in resource-constrained environments.

Strengths:
Performance Improvements: The proposed formulation achieves significant speedups—4x on CPUs and 9x on GPUs—for FFF depths up to 8 and 13, respectively. It also improves accuracy on CIFAR-10 by nearly 10%.

Unified Perspective: The matrix formulation provides a unified view of FFF and MoE, enabling better theoretical understanding and practical optimization.

Activation Function Exploration: The paper investigates the impact of different activation functions (Softplus, ReLU, linear) on FFF performance, revealing that simpler activations (e.g., linear) can outperform traditional choices.

Weaknesses:

Template and Rules:
Only 5 pages of the main text.

Limited Dataset: Experiments are conducted only on CIFAR-10, a lightweight dataset. While this allows for rapid iteration, it raises questions about the generalizability of results to more complex tasks and datasets. Notably, CIFAR-10 is typically solved with high accuracy using convolutional networks, which are not compared in this work.

Lack of Depth Analysis: The paper focuses on FFF depths up to 8 and 13 but does not provide a thorough analysis of how intermediate depths affect performance and accuracy.

Insufficient Justification for Activation Choices: While linear and ReLU activations show better results, the paper lacks a detailed explanation of why this occurs, particularly in the context of gradient propagation and model convergence. No discusinons and comapre with modern SOTA activations like SwiGLU and etc.

Memory and Parallelization Challenges: The use of sparse matrices is mentioned as a solution for memory constraints, but the paper does not discuss the computational overhead or implementation complexity associated with these structures.

Major Concerns:
Comparison with Full Architectures: The paper does not compare FFF or MoE with state-of-the-art architectureson CIFAR-10 or other datasets. This limits the practical relevance of the results, as CIFAR-10 is already solved with high accuracy using convolutional approaches. The study doesnt consider modern architectures used to solve practical tasks (transformers or earlier RNN or Resnet architectures and other more complex architectures)

GPU Performance Gap: Despite significant improvements, FFF remains 6.4x slower than MoE on GPUs. The paper does not explain why MoE retains such a strong advantage, nor does it propose solutions to close this gap.

Conclusion:
The paper presents an interesting and theoretically sound approach to unifying FFF and MoE, with notable improvements in speed and accuracy. However, the limited scope of experiments and lack of comparison with full-scale architectures reduce its practical impact.

---

### Official Review · Reviewer_1F3i · 2025-02-27
**A paper generalizing MoE and FFF but the experimental setting doesn't evaluate the full strengths of both architectures**

**Rating:** 7
**Confidence:** 4

**Review:**

This paper analyzes approaches to improving inference times in neural networks by means of selecting subnetworks for inference based on learned. The text is well-written and presents a clear description of the approach. The introduced general formula in a matrix form unifies binary tree-based FFF and MoE, which employs single-level routing.

## Weaknesses

The experimental setting doesn't evaluate the full strengths of both architectures.

- Sparsity penalties/load-balancing losses are not introduced, so the FFF model isn't encouraged to learn such a way that the applied greedy inference yields good results and prevent catastrophic forgetting.
- The dataset choice is questionable since MoE/FFF are expected to benefit from datasets composed of diverse, loosely connected samples (enabling the experts to specialize on various partitions of the dataset). At the same time, CIFAR-10 is believed to be classified based on generic image recognition patterns. Also the dataset size is small, and experiments with only one dataset are presented.
- Fully connected layers are used for image classification task, which is known to be suboptimal.

Also, it's not totally accurate to claim a higher speed of one architecture than another whereas their qualities or losses aren't matched. It is fairer to compare model performances with roughly the same numbers of parameters/FLOPs or, conversely, performances of models with comparable qualities. Hence, such a comparative study would be useful.

---

### Official Review · Reviewer_vFnC · 2025-02-28
**The paper aims to show more speed but lacks the generalization of proposed approach.**

**Rating:** 7
**Confidence:** 4

**Review:**

This paper examines methods to improve inference times in neural networks by selecting subnetworks based on learned criteria. The manuscript is structured clearly and outlines the approach with a general formula in matrix form that aims to unify the binary tree-based FFF with MoE, which uses single-level routing.

Advantages:

1. Clear experiment results

2. Decent explanation of problematic.

Disadvantages:

1. Lack of comparison with non-FFF architectures

2. Chosen data lacks diversity. It would be more interesting to see performance of that architecture in more complicated and heterogeneous data.

---

### Decision · Program_Chairs · 2025-03-08

**Decision:**

Accept (Oral)

**Comment:**

Your article has been accepted and you can make a presentation on the article. All articles will be sorted by rating and within the available conference places one author from each article will be invited. If there are not enough places, then you will either have the opportunity to present remotely or come at your own expense!